# Adhering to the 2017 Dutch Physical Activity Guidelines: A Trend over Time 2001–2018

**DOI:** 10.3390/ijerph17030681

**Published:** 2020-01-21

**Authors:** Marjolein Duijvestijn, Saskia W. van den Berg, G. C. Wanda Wendel-Vos

**Affiliations:** National Institute for Public Health and the Environment (RIVM), P.O. Box 1, 3720 BA Bilthoven, The Netherlands; saskia.van.den.berg@rivm.nl (S.W.v.d.B.); wanda.vos@rivm.nl (G.C.W.W.-V.)

**Keywords:** physical activity, guideline, exercise, health, trend, bone and muscle strengthening activities, cross-sectional

## Abstract

Recently, new physical activity (PA) guidelines were adopted in the Netherlands consisting of two components: (1) addressing duration of moderate and vigorous PA, (2) bone and muscle strengthening activities. The aim of this study is to retrospectively assess the long-term trend in fulfilling the criteria of the new PA guidelines and to gain insight into which activities contribute to changes over time. Data were available for 2001–2018 of a nationally representative sample of approximately 7000 Dutch citizens aged 12 years and over using the Short Questionnaire to Assess Health-enhancing physical activity (SQUASH). Multiple logistic regression analysis was performed by age, sex, and level of education. Overall, a positive trend was found from 39.9% adherence in 2001 to 46.0% in 2018. Adherence levels among adolescents decreased and increased among adults and seniors. Intermediate and higher educated groups showed positive trends over time whereas a stable trend was observed among lower educated. Activities contributing most to changes over time were sports, leisure time walking, and strenuous occupational activities. In the period 2001–2018, though an increasing trend was found, less than half of the population was sufficiently active. Special effort is necessary to reach adolescents, seniors, and lower educated groups in PA promotion programs.

## 1. Introduction

Non-communicable diseases (NCD) are a major health problem that could be countered by the elimination of physical inactivity [1]. In 2016, 28% of the world population was considered inactive, with a prevalence twice as high in Western countries [2]. It was estimated that 6%–10% of the burden of disease from NCD is attributable to physical inactivity [3]. In addition, the economic burden of physical inactivity on global health care costs has been conservatively estimated at 54 billion dollars in 2013 [4]. Insight into levels of physical activity and the trend over time, including preferences for certain types of activity among particular population groups is of crucial importance for policy makers [5].

In 2018, the World Health Organization (WHO) launched the global action plan on physical activity with the aim to help countries to scale up policy actions to promote physical activity [6]. The action plan provides a framework of effective and feasible policy actions at all levels. The goals set by the WHO for reducing physical inactivity are a relative reduction of 10% by 2025 and 15% by the year 2030 [6]. WHO has identified ten key areas for supporting countries in reaching these goals, one of which is to ‘monitor progress and impact of physical activity’ [6]. Currently in the Netherlands, three policies focus on sport and physical activity. First, the national sport agreement aims to preserve and strengthen the sport infrastructure, ensuring a safe environment for everyone to have fun and enjoy physical activity and sport [7]. Second, the cycling agenda comprehends increasing cycling kilometers with 20% (2017–2027) and to motivate an extra 200,000 employees to take up cycling to work [8]. Third, the Prevention Agreement states, among others, that adherence to the Dutch physical activity guidelines should increase to a level of 75% of the general population in 2040 as opposed to 47% in the year 2017 [9].

The current Dutch physical activity guidelines were adopted in 2017 as a result of an advisory report from the Dutch Health Council [10,11]. Table 1 shows the different elements of these guidelines.

This article will describe the second and third element: the time spent on physical activity of at least moderate intensity and the frequency of activities that strengthen muscles and bones. Both these components must be met to adhere to the physical activity guidelines. Compared to the former Dutch physical activity guidelines, the second component addressing bone and muscle strengthening activities is newly added [12]. To evaluate goals on adherence to the physical activity guidelines, monitoring and analysing trends over time are necessary [13,14,15].

Multiple studies have reported physical (in) activity levels at country [16,17,18,19,20,21], European [15,22], and/or global level [2,14,23]. A number of studies have also considered the development over time [2,18,19,20,24]. All studies included a national representative sample and reported separately on sex [2], age [18,19,20,24], and several on level of education as well [19,20,24]. However, these studies often only assessed a few time points over a period of less than 15 years [18,19,24]. A study by Guthold et al. (2018) was the first to assess a global long-term trend (2001–2016) of physical inactivity, distinguishing sex and income level [2]. They had to make many assumptions due to the use of several different questionnaires and only 55% of countries having data available for more than one time point [2]. In our study, annual data were available from 2001 to 2018. During this period, the same questionnaire was used. The large sample size of the dataset enables a selection for subgroups. Furthermore, the questionnaire items allow us to gain insight in the relative contribution of specific activities. To our knowledge, only few studies had this opportunity [25]. As far as we know, we are the first to combine yearly time points to assess long-term trends in adhering to a physical activity guideline and addressing activities that underlie changes over time.

In other countries, the bone and muscle strengthening component of a guideline rarely is measured in national surveillance systems and therefore most of the time not considered when calculating national prevalence estimates [26]. One study in Scotland has made an effort to provide detailed national representative information on the proportions meeting muscle strengthening guidelines [25]. They assessed the prevalence of muscle strengthening guidelines and participation in muscle strengthening guideline-specific exercises [25]. However, they did not integrate this measure with moderate to vigorous physical activity (MVPA). Another national study in Australia and the United States did examine incorporating muscle strengthening activities into their physical activity guideline. This resulted in lower adherence levels compared to the guideline of only MVPA [16,21]. To date, only limited knowledge is available concerning the attribution of bone and muscle strengthening exercises, to a guideline consisting of a combination of MVPA and bone and muscle strengthening exercises.

The overall aim of the present study is to gain more insight into adherence to the Dutch physical activity guidelines and its separate components addressing moderate and highly intense physical activity, and bone and muscle strengthening activities. Doing so by assessing the trend over time in the prevalence of adhering to the Dutch physical activity guidelines and underlying components. Additionally, to identify underlying physical activities that may be responsible for changes over time observed in adherence to the guidelines.

## 2. Materials and Methods

### 2.1. Study Design and Participants

From the year 2001, physical activity levels of the Dutch population were assessed on a yearly basis as part of a cross-sectional survey called the Dutch Health Survey/Lifestyle Monitor by Statistics Netherlands in collaboration with the National Institute for Public Health and the Environment [27,28,29,30]. In this survey, a two-stage stratified sampling design (1. municipalities and 2. individuals) was used to select participants from the Dutch Personal Records Database [31]. The sample was spread out over all months of the year [29]. Participants were requested to fill in the survey on paper or online, non-responders were approached face-to-face or by phone for an interview [27,28,29]. After weighting based on demographic characteristics such as age, sex, and marital status, the data are nationally representative of the Dutch population [32]. For comparability reasons, the present study used data from citizens aged 12 years and older, a sample of approximately 7000 Dutch citizens each year. Approval of the medical ethical committee was not necessary.

### 2.2. Demographic Characteristics

Participant characteristics (age, sex, level of education) were derived from the survey. Age groups were defined as adolescents (12 to 17 year old’s), adults (18 to 64 year old’s), and seniors (65 years and over). Highest attained level of education was classified as low (no, elementary, or low vocational/secondary schooling), intermediate (intermediate vocational or intermediate/higher secondary schooling), or high (higher vocational schooling or university) from the age of 25. Although information on the attained education was available bellow the age of 25, it was considered that most people would not have attained their highest level of education yet.

### 2.3. Physical Activity

Physical activity levels were assessed with the validated Short Questionnaire to Assess Health-enhancing physical activity (SQUASH) [33]. The SQUASH has been validated for adults (r = 0.43) [33], adolescents (r = 0.50) [34], older adults (r = 0.48) [35], and patient groups (r = 0.67) [36] by using double labelled water [34], accelerometer data [33,36], or another physical activity questionnaire [35]. The SQUASH domains and activities are presented in Table 2 [33]. Based on these items, using a standardized algorithm, adherence to the Dutch physical activity guidelines (Table 1) was calculated. In the algorithm, activities and sports were categorized as low (<3.0 MET) or moderate to vigorous intensity (≥3.0 MET) based on metabolic equivalent (MET) scores [37]. Based on the available data and the definition of the physical activity guidelines by the Dutch Health Council, activities and sports were categorized as muscle and/or bone strengthening [10,11]. Bone strengthening activities were defined as activities involving strength training and bearing body weight, for example, jumping, walking stairs, walking, running, and dancing [10,11]. Muscle strengthening activities included activities to increase strength, capacity, endurance, and muscle size, for example, exercise with the use of bodyweight and aerobic activities. Aerobic activities should involve large muscle groups, for example, walking, swimming, cycling, and dancing [11]. The categorization was reviewed by an expert group. Appendix A gives an overview of the MET scores corresponding to the activities and the categorization for bone and muscle strengthening activities.

### 2.4. Data Analysis

General population characteristics for each year were described (age, sex, level of education). Weighted prevalence were calculated for (i) meeting the full physical activity guidelines, (ii) meeting the MVPA component, (iii) meeting the bone and muscle strengthening component.

To analyze the trend over time, a logistic regression was performed with adhering to the overall guidelines or component as the dependent variable, resulting in three separate models. Time, the year of the measurement, was added to the model as a continuous independent variable. To investigate interaction by sex, age (in categories), and level of education (in categories) interaction terms were added one by one to the model. Data from 124,823 participants of 12 years and older were available for analysis of which 101,260 participants were from the age of 25. When the terms were statistically significant, stratified analysis was performed for subgroups. Adjusted odds ratios (ORs) and their 95% confidence intervals (CIs) were presented with three decimals, as ORs were interpreted per year, thereby expecting small ratios.

To get insight in the underlying activities that could explain observed trends, weighted mean time spent on activities (hours/week) was further investigated. The mean time spent on activities was assumed to represent duration and frequency and was presented for relevant subgroups (age, sex, level of education). Difference between years in time spent on activities of at least 0.5 h per week were considered to be relevant. Statistical analysis was conducted using IBM SPSS 22.0 statistical software (SPSS Inc. an IBM Company, Chicago, IL, USA). For all statistical tests, a *p*-value of <0.01 was used to indicate statistical significance.

## 3. Results

Table 3 shows the population size, general characteristics for all years, and the prevalence rates for 2018. The Dutch population distribution changed between 2001 and 2018, this is represented in our study population. For example, the proportion of seniors increased over the years and the proportion of lower educated declined.

In 2018, 46.0% of the total population aged 12 years and over adhered to the physical activity guidelines, 48.1% males and 43.9% females (Table 3). Adults adhered the most (50.1%), followed by seniors (37.0%) and adolescents (33.9%). For level of education, the adherence levels increased with a higher level of education (low (34.3%), intermediate (45.5%), and high (56.5%)).

In general, adherence levels for component 2, bone and muscle strengthening activities (78.2% in 2018), are higher compared to component 1, time spent on moderate to vigorous activities (52.4% in 2018, Table 3). In that sense, component 1 determines the level of adherence to the physical activity guidelines for the larger part. However, still a proportion of the individuals does adhere to component 1 and not to component 2 as adherence to the overall guidelines (46.0% in 2018) is lower compared to component 1 (52.4% in 2018).

Between 2001 and 2018, adherence to the Dutch physical activity guidelines increased from 39.9% to 46.0%, showing a stable to slightly increasing pattern (Figure 1a). This trend was similar for men and women (*p* > 0.01). For both adherence to the overall physical activity guidelines and the separate components, an interaction with age group was observed (*p* < 0.01). All prevalence data (2001–2018) can be found in Appendix B.

Among adolescents, a small negative trend over time was found with an odds for adherence to the overall guidelines of 0.979 [95% confidence interval (CI) 0.972:0.986] per year between 2001 and 2018 (34.3% and 33.9%). A similar trend was seen for component 1 (*p* < 0.01), whereas no trend was found for component 2 (*p* = 0.27, Figure 1). The decreasing trend may be explained by changes in time spent on specific activities (Figure 2a). Time spent on sports showed the largest difference between 2001 (i.e., 5.7 h/week) and 2018 (i.e., 4.6 h/week). Also, time spent on cycling to school/work clearly decreased during this period (4.2 vs. 3.6 h/week). On the other hand, an increase was seen in time spent on walking to work/school (0.7 vs. 1.3 h/week) and during leisure time (1.0 vs. 2.1 h/week).

For adults, a small positive trend over time was observed (OR 1.017 per year [95% CI 1.014:1.019]) in adherence to the overall guidelines between 2001 and 2018 (44.2% and 50.1%, Figure 1a). Similar trends were seen for component 1 (*p* < 0.01, Figure 1b) and component 2 (*p* < 0.01, Figure 1c). Subsequently, adults reported more time spent on several activities between 2001 and 2018 (Figure 2b). More time was spent on strenuous activity at work (4.5 vs. 6.2 h/week), sports (1.9 vs. 2.6 h/week), and leisure time walking (2.2 vs. 3.1 h/week, Figure 2b). Gardening is the only activity for which less time was spent in 2018 compared to 2001 (1.4 vs. 0.9 h/week).

Among seniors, a positive trend in adherence to the overall guidelines (OR 1.047 per year [95% CI 1.041:1.052]) was found as well, which was more pronounced than among adults (Figure 1a). Adherence to the physical activity guidelines among seniors increased from 22.0% in 2001 to 37.0% in 2018. The two underlying components showed the same increasing pattern during this period (*p* < 0.01, Figure 1b,c). The increasing trend may be explained by increasing amounts of time spent on walking as a leisure activity (2.7 vs. 4.0 h/week), sports (1.0 vs. 2.0 h/week), and strenuous household activities (1.0 vs. 1.6 h/week, Figure 2c) between 2001 and 2018.

Besides an interaction with age, also an interaction with level of education was observed (*p* < 0.01). A small positive significant trend over time in adherence to the physical activity guidelines was found for intermediate (OR 1.006 per year [95% CI 1.003:1.010]) and higher educated (OR 1.028 per year [95% CI 1.024:1.033]). Whereas, a stable trend over time was observed for lower educated (*p* = 0.97, 2001: 33.1%, 2018: 34.3%). Adherence to the overall guidelines increased for intermediate educated from 43.3% to 45.5% and for higher educated from 44.0% to 56.5% (Figure 1d). Similar trends were seen for component 1 and 2 (*p* < 0.01). Over the years (2001 vs. 2018), intermediate educated showed an increase in time spent on strenuous activities at work (4.9 vs. 6.2 h/week) and leisure time walking (2.4 vs. 3.4 h/week, Figure 3b). While, they spent less time on gardening (1.9 vs. 1.3 h/week). Higher educated showed an increase in time spent on sports (2.0 vs. 3.2 h/week), strenuous activities at work (1.0 vs. 2.8 h/week), and leisure time walking (2.1 vs. 3.1 h/week, Figure 3c).

## 4. Discussion

In the present study, we found that from 2001 to 2018, adherence to the Dutch physical activity guidelines increased for adults, seniors, and the intermediate and higher educated. An inverse trend was found for adolescents and a stable trend was found for the lower educated. Similar trends were seen for the two underlying components. Generally, adults adhered the most to the physical activity guidelines and seniors the least. Although, seniors showed the largest improvement over time. The underlying physical activity pattern for the trend over time of the components differed for each age and level of education group. In general, the activities that contributed most to changes in adhering to the Dutch physical activity guidelines were strenuous activities at work, sports, and walking during leisure time.

In 2018, less than half of the Dutch population over 12 years old adheres to the physical activity guidelines (46.0%). Several other countries have reported adherence levels on their physical activity guidelines; 65.2% of American adults [19], 45% of Australian adults [38], and 66% of English males and 58% of English female aged 16 year and older [39]. However, for methodological reasons, comparisons of adherence rates between countries must be treated with caution. First of all, these countries use different definitions of being sufficient physical active compared to the Netherlands and each other. Moreover, various validated and unvalidated questionnaires exist and are used by countries to assess physical activity levels, prohibiting clear comparison between countries [40]. Therefore, differences in methodology should be taken into account when comparing results between studies and countries. Nonetheless, it is of interest to examine relative differences between groups (e.g., sex, age), the trend over time, and share these outcomes between countries.

The Dutch physical activity guidelines consist of a moderate to vigorous physical activity (MVPA) and a bone and muscle strengthening component. A few studies assessed muscle strengthening activity in a physical activity guideline [16,21]. A recent national American study by Bennie et al. (2019) found that by adding muscle strengthening activities to an MVPA guideline, resulted in lower adherence levels (30.5% vs. 20.3%) [21]. The results are in line with our findings, though in our study, a smaller decline was found by the addition of a bone and muscle strengthening component, from 52.4% to 46.0%. Bennie et al. (2019) used a different question to assess muscle strengthening activities. In this question, they explicitly excluded walking, running, and cycling. While in our study, these activities are seen as muscle strengthening based on a report by the Committee for Guidelines on Physical Activity [11]. Consequently, methodological differences can explain the different findings.

Strengths include the use of a large representative study population, giving us the possibility to investigate several subgroups. Data were available for each year from 2001 to 2018, with no missing data. Furthermore, the adherence to the physical activity guidelines was assessed retrospectively with the calculation method being the same for all years. Moreover, our findings give insight into which activities the population performs to adhere to the Dutch physical activity guidelines.

Despite the questionnaire being the same over time, data collection was subjected to methodological changes, which counts as a limitation of our study. Until 2013, data were collected by interviews and on paper. From 2014 and on, it was also possible to fill in the survey online. A report by Statistics Netherlands showed differences for a selection of variables due to this change [41]. The effect of this change has not been tested specifically for physical activity variables. In our results, no sudden changes were observed over all adherence levels to the physical activity guideline in 2014. Next to that, in essence, the questionnaire has been the same for all years, only a few years had small technical differences. For a few questions, technical differences involved the possibility to skip a question when not applicable to the participants situation. Despite small changes in the sample and collection method, it can be concluded that changes in methodology had a negligible effect on the results of our study.

Another limitation of our study includes the reliance on self-reported physical activity (e.g., overestimation, social desirable answers, recall bias). Specific limitation of the SQUASH questionnaire is an overestimation of physical activity due to the large number of items compared to other questionnaires. However, since 2001, the physical activity questionnaire has been the same in our study. Therefore, the bias related to overestimation may be considered a constant factor over time and less of a factor to consider when studying trends in physical activity levels. Therefore, it seems less likely that overestimation of physical activity levels has affected our trend conclusions. To overcome bias related to self-report, objective measurements could be used [42]. Using accelerometers in national surveillance systems has potential, however, more research on data quality and practical implications is necessary. For example, some activities may be poorly represented in accelerometer data. When assessing adherence to physical activity guidelines there might be a problem with including bone and muscle strengthening activities. Also, in countries where cycling is an important activity, such as the Netherlands, the total amount of physical activity may contain a systematic error. We also have to consider the fact that current physical activity guidelines and recommendations are largely based on self-reported studies using questionnaires when examining associations between activity and health-related outcomes [43]. Recommendations may change based on future large-scale longitudinal studies using objective measurements related to health-related outcomes [43]. Consequently, using objective measures in national surveillance systems aiming to assess physical activity levels related to good/better health may entail using a different indicator for health enhancing physical activity.

In our study, strenuous activities at work were found to attribute to an increasing adherence to the physical activity guidelines over time. However, it should be noted that previous research has indicated that occupational physical activity elevates cardiovascular risk [44]. In our article, strenuous activities at work were seen as activities to adhere to the physical activity guidelines, as the guidelines do not make a distinction where activities have taken place [10,11]. However, bias caused by including occupational activities in the analysis might be less pronounced as our population is national representative with only a portion in strenuous labor force. Nonetheless, occupational physical activities should not be seen as activities to be promoted for health-enhancing purposes [44].

In our study, adherence levels to the Dutch physical activity guidelines were found to differ between population groups. In general, physical activity levels tend to differ between men and women [2,18,45]. In the Netherlands, adherence levels are comparable between men and women. Another interesting finding of our study is that, seniors became more active. At the same time, physical activity levels are still relatively low compared to other age groups. Since, staying physically active is of tremendous importance to live independently as long as possible [46]. It is important to find ways to even further increase physical activity levels among seniors in the future. Opposite to seniors, adolescents became less active in the period 2001–2018. This is of concern because literature suggested that physical activity levels in childhood can track to adulthood [47,48]. This makes it of even more importance to facilitate physical activity among adolescent groups.

Next to developments within and between age groups, our results show an increasing gap of physical activity levels between lower educated at the one side and higher educated at the other. The widening of the gap between higher and lower educated is well known for health status [49,50,51,52]. Also, we are not the first to confirm this for physical activity [50,51]. This emphasizes the need for physical activity promotion tailored to lower educational levels in order tackle these social economic differences in physical activity level.

It is likely that more population groups stay behind with respect to adherence to physical activity guidelines such as people with disabilities, overweight, or living in urban or rural areas. Future studies can contribute by providing more insight in such groups. It would also be interesting to answer questions on additional time needed to change non-adherence to adherence. In that respect, it may be helpful to take a closer look at specific activities that largely contribute to adherence such as specific sports or walking for particular purposes. It is advised to perform analysis separately for the two components, as some individuals already adhere to one component and not yet to the other. The outcome of such future studies on other relevant subgroups and specific activities could help tailoring physical activity promotion to certain target populations.

Overall, a small positive trend is observed from 2001 to 2018 for the Dutch population of 12 years and older. Still, adolescents, seniors, and lower educated groups are in need of focused attention to improve their adherence rates. Physical activity levels could be improved by policy changes aiming at tailoring physical activity promotion programs to subgroups [6]. To take a step in shaping these physical activity promotion programs, future studies could investigate which specific activities should be stimulated by policy actions to increase the physical activity levels.

From 2001 to 2018, adherence rates by the Dutch population increased from 39.9% to 46.0%. The Dutch government has the ambitious goal to have 75% of the population adhere to the physical activity guidelines by 2040 [9]. The goal set by the WHO might by more realistic with a relative increase of 15% in 2030 compared to 2018 [6]. Meaning, for the Dutch situation to reach 53% adherence in 2030. Still, large measures and policy actions are necessary to accomplish either goals. Collaborations between sectors (e.g., health care, education, communities, and sport clubs) is of importance to increase population adherence rates and to target the groups with the lowest adherence levels [6].

## 5. Conclusions

Adherence levels for the Dutch physical activity guidelines have slightly increased between 2001 and 2018. Activities that contributed to changes in adhering to the Dutch physical activity guidelines were strenuous activities at work, sports, and walking during leisure time. Still, less than half of the population is sufficiently active. Especially adolescents, seniors, and groups with a lower level of education are in need of attention for tailored physical activity programs. In collaboration with multiple stakeholders actions can be taken to accomplish the adherence goals set out by the Dutch government and the WHO.

## Figures and Tables

**Figure 1 ijerph-17-00681-f001:**
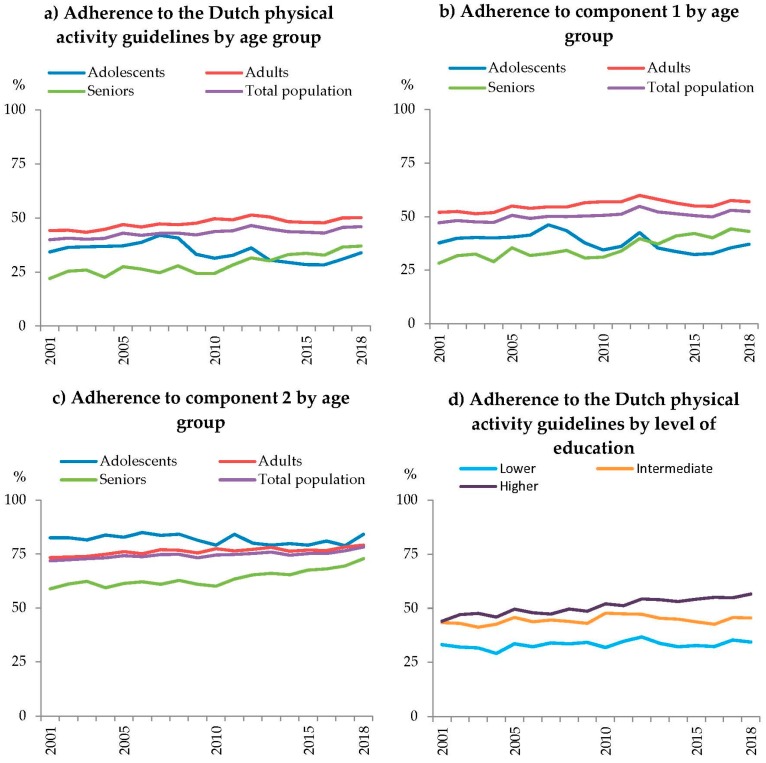
Adherence levels to the Dutch physical activity guidelines (%) consisting of component 1; 150 moderate-and vigorous intense physical activity per week for adults, one hour per day for adolescents and component 2; two times per week bone and muscle strengthening activities for adults, three times for adolescents, stratified by age groups (12–17, 18–64, 65+) and level of education. (**a**) Adherence to guidelines for age groups, (**b**) adherence to component 1 for age groups, (**c**) adherence to component 2 for age groups, (**d**) adherence to the guidelines by level of education.

**Figure 2 ijerph-17-00681-f002:**
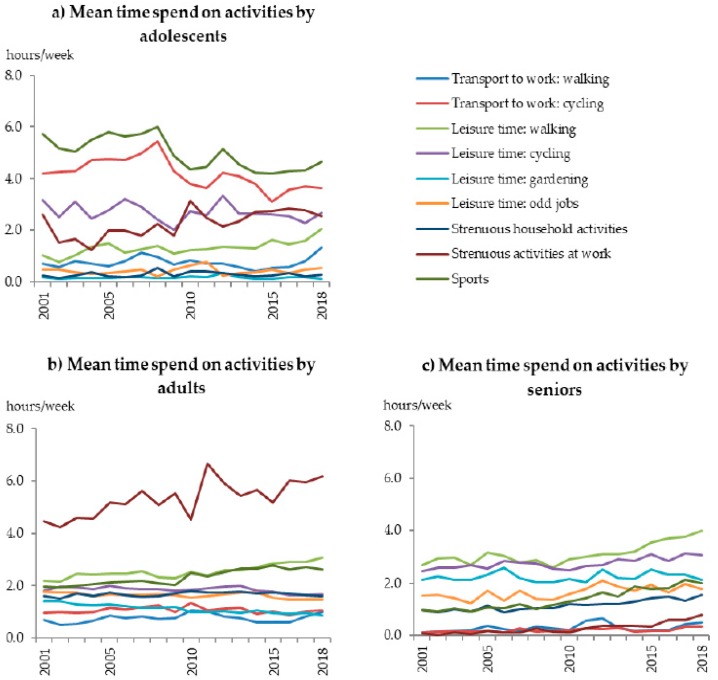
Mean time spend on activities at work or school, in household, during leisure time and sports per week for each year (2001–2018) stratified for age groups, (**a**) adolescents (12–17 years old), (**b**) adults (18–64 years old), (**c**) seniors (65+ years old).

**Figure 3 ijerph-17-00681-f003:**
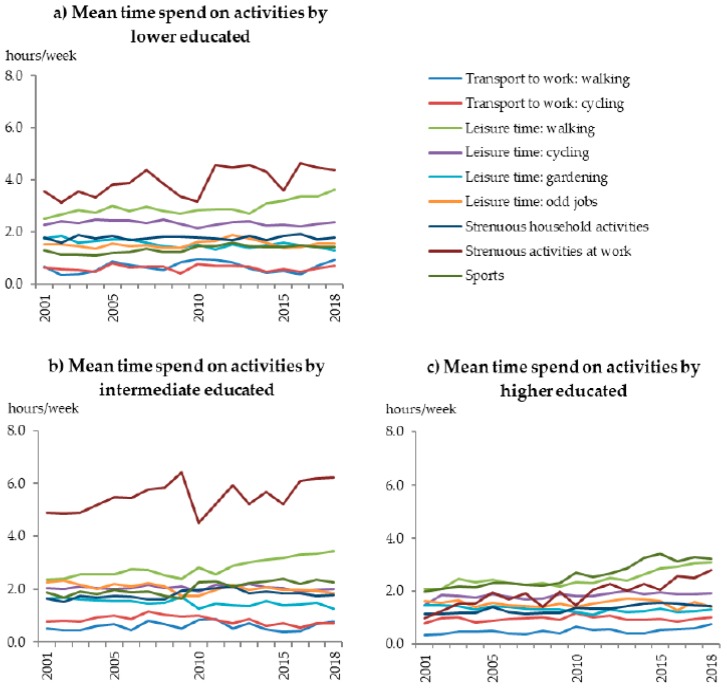
Mean time spend on activities at work or school, in household, during leisure time and sports per week for each year stratified for level of education, (**a**) lower educated (no, elementary, or low vocational/secondary schooling), (**b**) intermediate educated (intermediate vocational or intermediate/higher secondary schooling), (**c**) higher educated (higher vocational schooling or university).

**Table 1 ijerph-17-00681-t001:** Dutch physical activity guidelines 2017 [10,11].

Adults and Older People ≥18 Years of Age	Children and Adolescents 4–18 Years of Age
Physical activity is good for you—the more, the better Engage in physical activity of moderate intensity for at least 150 min every week, spread over several different days. For example, walking and cycling. The longer you are physically active, and the more frequent and/or more vigorous the activity, the more your health will benefit. Do activities that strengthen your muscles and bones at least twice a week. Older people should combine these with balance exercises. And: avoid spending long periods sitting down.	Physical activity is good for you—the more, the better Engage in physical activity of moderate intensity for at least one hour every day. The longer you are physically active, and the more frequent and/or more vigorous the activity, the more your health will benefit. Do activities that strengthen your muscles and bones at least twice a week. Older people should combine these with balance exercises. And: avoid spending long periods sitting down.

**Table 2 ijerph-17-00681-t002:** Domains and activities of the Short Questionnaire to Assess Health-enhancing physical activity (SQUASH) [33].

Domain	Activities ^†^	Domain	Activities ^†^
Transport to work/school	Walking to work/school ^†^ Cycling to work/school ^†^	At work/school	Working activities, light/moderate * Working activities, strenuous *
Leisure time	Walking in leisure time ^†^ Cycling in leisure time ^†^ Gardening ^†^ Odd jobs ^†^	Household	Household activities, light/moderate ^†^ Household activities, strenuous ^†^
		Sports	Sports (max. 4) ^†^

^†^ assessed in number of days per week and hour/minutes on those days. * assessed in hours per week and for intensity: light/moderate activities at work were defined as sitting/standing work with occasional walking such as office work or walking during work with light loads, strenuous activities were defined as walking during work or work for which heavy loads must be lifted regularly.

**Table 3 ijerph-17-00681-t003:** Population size (*n*) for each year included in the analysis stratified by age, sex, and by level of education (#%) and the prevalence of adherence to the Dutch physical activity guidelines and components: (1) moderate and vigorous intense physical activity, (2) bone and muscle strengthening activities, for all subgroups (#,#%).

	Total (*n*)		Age Group (%) *			Sex (%) *		Level of Education (%) ^†,^ᵟ		
Year	≥12 Years	≥25 Years	Adolescents (12–17)	Adults (18–64)	Seniors (65+)	Male	Female	Lower	Intermediate	Higher
2001	5971	4915	9	75	16	48	52	43	34	23
2002	5834	4775	9	74	17	47	53	42	34	24
2003	6566	5409	10	74	16	48	52	41	34	25
2004	7584	6157	10	74	16	48	52	31	39	30
2005	7128	5930	9	73	18	48	52	42	34	24
2006	6733	5601	9	73	18	48	52	41	34	25
2007	5951	4953	9	72	19	47	53	41	33	26
2008	6212	5170	9	72	20	48	52	39	34	27
2009	5962	3248	8	71	21	48	52	37	34	29
2010	7218	6085	7	72	21	47	53	38	30	32
2011	6262	5238	7	71	22	48	52	39	29	32
2012	6349	6085	7	71	22	47	53	34	33	34
2013	6444	5241	7	70	23	46	54	26	41	33
2014	7859	6331	9	69	23	51	49	30	40	30
2015	7783	6272	9	69	22	52	48	30	40	30
2016	7646	6137	9	68	23	48	52	30	38	31
2017	8819	6598	9	66	25	49	51	28	40	32
2018	8502	7115	8	70	22	49	51	26	39	35
Prevalence rates for adherence to the physical activity guidelines in 2018 (#,#%)										
Overall	46.0	46.2	33.9	50.1	37.0	48.1	43.9	34.3	45.5	56.5
Component 1	52.4	52.8	37.1	56.9	43.1	55.7	49.1	42.5	52.8	61.3
Component 2	78.2	77.6	84.2	79.1	72.9	78.2	78.3	67.4	77.3	86.3

* Based on population size of ≥12 years old ^†^ Based on population size of ≥25 years old and categorized by highest attained level of education ^ᵟ^ low (no, elementary, or low vocational/secondary schooling), intermediate (intermediate vocational or intermediate/higher secondary schooling), or high (higher vocational schooling or university).

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
