# Peer review of "Adhering to the 2017 Dutch Physical Activity Guidelines: A Trend over Time 2001–2018"

_ijerph, 2020, doi:10.3390/ijerph17030681_

Round 1

Reviewer 1 Report

Dear authors:

The article proposes an interesting topic and presents all the parts that are required for this type of articles. However, it is recommended to pay attention to the following points in order to improve the article:

1. The meaning of acronyms should be given the first time they are quoted. For example, WHO.
2. In Figure 1 the reference to graphic "b" should be revised as it is referenced as "c".
3. The results should indicate the % explanation of the logistic regression models. It should also be indicated whether the fit of the model has been checked, e.g. through Nagelkerke R2. A table should also be included showing all OR coefficients, their intervals and the associated probability of each independent variable entered in the models.
4. It is recommended in the final part of the discussion to further detail the practical implications. Possible actions in sports policies that could be implemented should be proposed.

Author Response

Response to Reviewer 1 Comments

Point 1: The article proposes an interesting topic and presents all the parts that are required for this type of articles. However, it is recommended to pay attention to the following points in order to improve the article

Response 1: We would like to thank the reviewer for evaluating our manuscript. We have tried to address all the reviewers’ concerns in a proper way. We would be happy to make further corrections if necessary.

Point 2: The meaning of acronyms should be given the first time they are quoted. For example, WHO.

 Response 2: We thank the reviewer for the thorough read. We have changed this as suggested.

Point 3: In Figure 1 the reference to graphic "b" should be revised as it is referenced as "c".

Response 3: We thank the reviewer for pointing out the reference mistake. We have changed this as suggested.

Point 4: The results should indicate the % explanation of the logistic regression models. It should also be indicated whether the fit of the model has been checked, e.g. through Nagelkerke R2. A table should also be included showing all OR coefficients, their intervals and the associated probability of each independent variable entered in the models.

Response 4: We thank the reviewer for this comment. We have taken this comment in consideration and verified it with our statistics department. The goal of the analysis was to examine whether a positive/negative/stable trend over time existed, based on a pre-defined set of independent variables (year, sex, age, level of education). Thereby the  % of variance explained and the goodness-of-fit of the model is less of an issue as we are not searching for the best explanatory model. The logistic regression model is only used to address whether a significant trend existed and its direction. Next to that, all OR coefficients (and confidence intervals) indicating trends are presented in the text with their interpretation. To not duplicate information we decided not to include a table with the OR coefficients.

Point 5: It is recommended in the final part of the discussion to further detail the practical implications. Possible actions in sports policies that could be implemented should be proposed.

Response 5: We thank the reviewer for this suggestion and agree that practical implications would be valuable for policy action. However, the scope of the article was to address retrospectively activities that were important for changes over time in adherence from an observational perspective. These findings cannot directly be translated to the future as our research is only observational.

To enable us to give practical implications for policy actions it is (indirectly) suggested that more research is necessary. Such as gaining insight into how much additional time is needed to change non-adherence to adherence (line 332-333) and which activities could be stimulated to increase physical activity levels (line 333-335). Next to that, at the end of the discussion it is suggested that collaboration between sectors is necessary to accomplish higher population adherence levels (line 350-352). We believe that the physical inactivity problem is not just a problem to solve for sports policy but for the health, education and commuting policymakers as well.

Reviewer 2 Report

This paper uses data from a nationally representative survey of adolescents and adults from the Netherlands to investigate trends in adherence to physical activity and muscle strengthening guidelines. The strength of the study is the use of a nationally representative sample and the longitudinal design with measures being undertaken from 2001-2018. The authors also attempt to establish adherence to the often "forgotten" muscle-strengthening guideline - although the method used to determine this requires more explanation.

My main concern is the lack of detail regarding the self-report instrument's validity and its limitations that may have influenced the trend results.

As the SQUASH measure is the primary data collection instrument, more detail regarding its previous validity testing is required (rather than just providing a citation). Please note what population was used in the validity trial (important as it was only n = 50 in the age range of 27-58? Therefore not validated in age groups younger or older?). What criterion measure was used to establish validity and state the validity statistic/result.

Further discussion of how activities were classified as muscle strengthening is required. Is the only defining characteristic "activities using large muscle groups"? What is the full definition in the [11] reference? Who made the classification decisions for activities reported in this study? The Appendix A table cites the Ainsworth compendium for MET-score, muscle-strengthening & bone-strengthening, yet the Ainsworth compendium only provides MET-scores

What definitions/examples are participants provided to help determine if their "working activities" are "vigorous"? As the PA guidelines use moderate intensity as their health-enhancing cut-point why were activities classified as "light/moderate" and "vigorous"? Why not "moderate/vigorous" which is more standard practice. Figure 2 refers to "strenuous" activities. Is this "vigorous"? Consistency in terminology would be helpful.

Why max 4 sports? How might these questionnaire design factors impact your findings? E.g. You found that adolescent sport participation declined, but could adolescents in 2018 be involved in greater varieties of sports (> 4) for shorter durations or less frequently each? (Therefore, total sport participation may not have changed so dramatically?).

Where are fitness/gym activities reported?

The discussion of self-report as a limitation at line 275 requires extra attention. Not all self-report questionnaires/approaches are equal and it would appear that the SQUASH questionnaire has a number of specific limitations related to the use of wide categories rather than specific activities, etc. A more thoughtful discussion of self-report limitations broadly and limitations specific to the SQUASH is recommended.

As work based PA is reported, consider comparing results to: Karin I. Proper, Vincent H. Hildebrandt,
Physical activity among Dutch workers—differences between occupations,
Preventive Medicine,
Volume 43, Issue 1, 2006

Did you consider seasonal effects?: Plasqui, G. and Westerterp, K.R. (2004), Seasonal Variation in Total Energy Expenditure and Physical Activity in Dutch Young Adults. Obesity Research, 12: 688-694. doi:10.1038/oby.2004.80

Editing:

A thorough proof-read is required throughout. Some notable edits required include:

Use "data were" rather than "data was" as data are plural;

Line 11 in Abstract: Add "The" so the sentence reads "The aim of this study is to..."

Line 114: assessed (not assed)

Author Response

Response to Reviewer 2 Comments

Point 1: This paper uses data from a nationally representative survey of adolescents and adults from the Netherlands to investigate trends in adherence to physical activity and muscle strengthening guidelines. The strength of the study is the use of a nationally representative sample and the longitudinal design with measures being undertaken from 2001-2018. The authors also attempt to establish adherence to the often "forgotten" muscle-strengthening guideline - although the method used to determine this requires more explanation.

Response 1: We would like to thank the reviewer for evaluating our manuscript. We have tried to address all the reviewers’ concerns in a proper way and believe that our paper has improved considerably. For the method issue on the muscle-strengthening guidelines we would like to refer to point 3.

Point 2: My main concern is the lack of detail regarding the self-report instrument's validity and its limitations that may have influenced the trend results. As the SQUASH measure is the primary data collection instrument, more detail regarding its previous validity testing is required (rather than just providing a citation). Please note what population was used in the validity trial (important as it was only n = 50 in the age range of 27-58? Therefore not validated in age groups younger or older?). What criterion measure was used to establish validity and state the validity statistic/result.

Response 2: We thank the reviewer for the suggestion and have added the following sentence in the method section, line 115-118 (new information is underlined):  

“Physical activity levels were assessed with the validated Short Questionnaire to Assess Health Enhancing Physical Activity (SQUASH) [33]. The SQUASH has been validated for adults (r=0.43)[33], adolescents (r=0.50) [34], older adults (r =0.48)[35] and patient groups (r=0.67)[36] by using double labelled water [34], accelerometer data[33,36] or another physical activity questionnaire[35].”

In the discussion we have also added extra information on the limitations of the SQUASH questionnaire, line 287-293 . It now reads (new information is underlined):

“Specific limitation of the SQUASH questionnaire is an overestimation of physical activity due to the large number of items compared to other questionnaires. However, since 2001 the physical activity questionnaire has been the same in our study. Therefore, the bias related to self-report overestimation may  be considered a constant factor over time and less of a factor to consider when studying trends in physical activity levels. Therefore, it seems less likely that overestimation of physical activity levels has affected our trend conclusions.”

 Point 3: Further discussion of how activities were classified as muscle strengthening is required. Is the only defining characteristic "activities using large muscle groups"? What is the full definition in the [11] reference? Who made the classification decisions for activities reported in this study? The Appendix A table cites the Ainsworth compendium for MET-score, muscle-strengthening & bone-strengthening, yet the Ainsworth compendium only provides MET-scores

Response 3: We agree that this point, highlighted by two Reviewers, is important to clarify. The national report that is referred to states the following definition for muscle strengthening activities: “activities to increase strength, capacity, endurance and muscle size, for example exercise with the use of bodyweight and aerobic activities such as cycling”. The aerobic activities have also been defined by the report as: “activities that aim to increase endurance for which large muscle groups are used, for example walking, swimming, cycling and dancing”. Based on this definition, activities were categorized by the researchers and verified with an expert group. We have added a few sentences to the method section ‘2.3 Physical Activity’, line 121-133. It now reads:

“In the algorithm, activities and sports were categorized as low (<3,0 MET) or moderate- to vigorous intensity (≥3,0 MET) based on Metabolic Equivalent (MET) scores [37]. Based on the available data and the definition of the physical activity guidelines by the Dutch health council, activities and sports were categorized as muscle- and/or bone strengthening [10,11]. Bone strengthening activities were defined as activities involving strength training and bearing body weight, for example jumping, walking stairs, walking, running and dancing bearing body weight [10,11]. Muscle strengthening activities included activities to increase strength, capacity, endurance and muscle size, for example exercise with the use of bodyweight and aerobic activities. Aerobic activities should involve large muscle groups, for example walking, swimming, cycling and dancing [11]. The categorization was reviewed by an expert group. Appendix 1 gives an overview of the MET scores corresponding to the activities and the categorisation for bone- and muscle strengthening activities.”

To Appendix A we added an extra reference (to the health council report and article) and added the following information (line 374-379):

“* Based on the available data and the definition of the physical activity guidelines by the Dutch Health Council activities and sports were categorized as either muscle- and/or bone strengthening. Bone strengthening activities were defined as activities involving strength training and bearing body weight, for example jumping, walking stairs, walking, running and dancing. Muscle strengthening activities included activities to increase strength, capacity, endurance and muscle size, for example exercise with the use of bodyweight and aerobic activities. Aerobic activities should involve large muscle groups, for example walking, swimming, cycling and dancing.”

 Point 4: What definitions/examples are participants provided to help determine if their "working activities" are "vigorous"? As the PA guidelines use moderate intensity as their health-enhancing cut-point why were activities classified as "light/moderate" and "vigorous"? Why not "moderate/vigorous" which is more standard practice. Figure 2 refers to "strenuous" activities. Is this "vigorous"? Consistency in terminology would be helpful.

Response 4: We thank the reviewer for this valuable suggestion and agree that the inconsistent use of words is confusing. In Figure 2 “vigorous” is supposed to be “strenuous”, this has been adjusted.

The definition of work intensity has been added to the asterix of Textbox 2, line 137-140. It now reads:

“  *assessed in hours per week and for intensity: light/moderate activities at work were defined as sitting/standing work with occasional walking such as office work or walking during work with light loads, strenuous activities were defined as walking during work or work for which heavy loads must be lifted regularly.”

From these examples it becomes clear that the definition of strenuous work activity is closely related to MVPA criteria and light/moderate activities at work as light physical activity.

 Point 5: Why max 4 sports? How might these questionnaire design factors impact your findings? E.g. You found that adolescent sport participation declined, but could adolescents in 2018 be involved in greater varieties of sports (> 4) for shorter durations or less frequently each? (Therefore, total sport participation may not have changed so dramatically?).

Response 5: The decision to ask for a maximum of 4 sports was arbitrary at first. We have found that in 2001 80% of the adolescents reported to participate in 1 sport, 47% in 2 sports, 17% in 3 and 6% in 4. For 2018 these numbers were respectively, 75%, 28%, 8% and 3%. Indeed we would miss it when someone participates in more sports. However, the low percentage for participating in 4 sports does indicate that this number will be small. Also, based on the decline in percentages over the years for all number of sports we argue that the variety of sports participated in has not increased.

 Point 6: Where are fitness/gym activities reported?

Response 6: Fitness and gym activities were reported in the sports question. Examples that were given in the question about sports were: “fitness/endurance training, tennis, running, football”.

Point 7: The discussion of self-report as a limitation at line 275 requires extra attention. Not all self-report questionnaires/approaches are equal and it would appear that the SQUASH questionnaire has a number of specific limitations related to the use of wide categories rather than specific activities, etc. A more thoughtful discussion of self-report limitations broadly and limitations specific to the SQUASH is recommended.

Response 7: A distinctive characteristic of the SQUASH questionnaire is the use of specific activities within activity domains. In total it includes 10-14 items depending on the number of sports that people engage in. This relative large number of items in the questionnaire feeds overestimation of activity levels. To mention this limitation a sentence has been added to the paragraph on self-report bias in the discussion. Also we added information to indicate other limitations of self-report. The paragraph now reads, new information is underlined, line 286-305:

“Another limitation of our study includes the reliance on self-reported physical activity (e.g. overestimation, social desirable answers, recall bias). Specific limitation of the SQUASH questionnaire is an overestimation of physical activity due to the large number of items compared to other questionnaires. However, since 2001 the physical activity questionnaire has been the same in our study. Therefore, the bias related to self-report overestimation may be considered a constant factor over time and less of a factor to consider when studying trends in physical activity levels. Therefore, it seems less likely that overestimation of physical activity levels has affected our trend conclusions. To overcome bias related to self-report, objective measurements could be used [42]. Using accelerometers in national surveillance systems has potential, however more research on data quality and practical implications is necessary. For example, some activities may be poorly represented in accelerometer data. When assessing adherence to physical activity guidelines there might be a problem with including bone- and muscle strengthening activities. Also, in countries where cycling is an important activity, such as the Netherlands, the total amount of physical activity may contain a systematic error. We also have to consider the fact that current physical activity guidelines and recommendations are largely based on self-reported studies using questionnaires when examining associations between activity and health-related outcomes [43]. Recommendations may change based on future large-scale longitudinal studies using objective measurements related to health-related outcomes [43]. Consequently, using objective measures in national surveillance systems aiming to assess physical activity levels related to good/ better health may entail using a different indicator for health enhancing physical activity.”

 Point 8: As work based PA is reported, consider comparing results to: Karin I. Proper, Vincent H. Hildebrandt, Physical activity among Dutch workers—differences between occupations,
Preventive Medicine, Volume 43, Issue 1, 2006

Response 8: We thank the reviewer for the suggestion and are familiar with the study by Proper & Hildebrandt (2006). They describe total physical activity and the contribution of work to total physical activity for various occupational groups. In their study, they only included a working population. In our study we did not report on this specific group. Also, the study was conducted in the years 2000-2002. During that time other physical activity guidelines were used and reported on. For these reasons our results cannot easily be compared to each other.  

Point 9: Did you consider seasonal effects?: Plasqui, G. and Westerterp, K.R. (2004), Seasonal Variation in Total Energy Expenditure and Physical Activity in Dutch Young Adults. Obesity Research, 12: 688-694. doi:10.1038/oby.2004.80

Response 9: The data collection was performed year round. Therefore we consider seasonal effects to be eliminated. See also method section line: 99-100:

“ The sample was spread out over all months of the year [29].”.

Point 10: Editing:

A thorough proof-read is required throughout. Some notable edits required include:

Use "data were" rather than "data was" as data are plural;

Line 11 in Abstract: Add "The" so the sentence reads "The aim of this study is to..."

Line 114: assessed (not assed)

Response 10: We thank the reviewer for thoroughly reading the manuscript. We have changed the text as suggested.

Reviewer 3 Report

The authors should be complemented for addressing this interesting topic. Apart from the observational and cross-sectional nature of the study, the large representative sample combined with a time trend analysis over a period of 17 years are important assets of the study. Nonetheless, the following comments should be taking into account in a revised version of the manuscript.

1. A main concern is the applied methodology/questions used to assess muscle- and bone-strengthening activities. As mentioned by the authors, previous studies applied different questions excluding aerobic activities such as walking and cycling. The authors refer to a national report on the physical activity guidelines 2017 to motivate their decision. Unfortunately, the provided web link in the reference list (11) does not work. Moreover, from appendix A it is also unclear how the specific sport activities were classified as muscle- or bone strengthening activities. I would therefore recommend elaborating on the applied approach to assess muscle- and bone-strengthening activities in the present study and to motivate their decision from a more international perspective. As cycling and walking was assessed separately, prevalence rates excluding aerobic activities such as walking and cycling as muscle-strengthening activities could be calculated.

2. It was concluded that changes in strenuous activities at work, sports and walking during leisure time contributed the most to observed time trends in adherence to the Dutch physical activity guideline. From appendix A, the estimated energy expenditure of walking during leisure time was 2.7 MET. Based on the physical activity guidelines, people should engage in physical activity of moderate-to-vigorous (MVPA) intensity (>3MET). Consequently, walking during leisure time (2.7 MET) should have been excluded in the algorithm for calculating MVPA. Furthermore, previous work on the ‘physical activity paradox’ argues that occupational physical activity is associated with an elevated cardiovascular risk (e.g., Holtermann et al., British Journal of Sports Medicine 2012;46:291-295). This could suggest that promoting occupation physical activity as health-enhancing is not recommended. Please comment on this in the discussion part of the paper.

Author Response

Response to Reviewer 3 Comments

Point 1: The authors should be complemented for addressing this interesting topic. Apart from the observational and cross-sectional nature of the study, the large representative sample combined with a time trend analysis over a period of 17 years are important assets of the study. Nonetheless, the following comments should be taking into account in a revised version of the manuscript.

Response 1: We would like to thank the reviewer for evaluating our manuscript. We have tried to address all the reviewers’ concerns in a proper way and believe that our paper has improved considerably. We would be happy to make further corrections if necessary.

Point 2: A main concern is the applied methodology/questions used to assess muscle- and bone-strengthening activities. As mentioned by the authors, previous studies applied different questions excluding aerobic activities such as walking and cycling. The authors refer to a national report on the physical activity guidelines 2017 to motivate their decision. Unfortunately, the provided web link in the reference list (11) does not work. Moreover, from appendix A it is also unclear how the specific sport activities were classified as muscle- or bone strengthening activities. I would therefore recommend elaborating on the applied approach to assess muscle- and bone-strengthening activities in the present study and to motivate their decision from a more international perspective. As cycling and walking was assessed separately, prevalence rates excluding aerobic activities such as walking and cycling as muscle-strengthening activities could be calculated.

Response 2: We agree that this point, highlighted by two Reviewers, is important to clarify and therefore have added a few sentences to the method section ‘2.3 Physical Activity’, line 121-133. It now reads:

“In the algorithm, activities and sports were categorized as low (<3,0 MET) or moderate- to vigorous intensity (≥3,0 MET) based on Metabolic Equivalent (MET) scores [37]. Based on the available data and the definition of the physical activity guidelines by the Dutch health council, activities and sports were categorized as muscle- and/or bone strengthening [10,11]. Bone strengthening activities were defined as activities involving strength training and bearing body weight, for example jumping, walking stairs, walking, running and dancing bearing body weight [10,11]. Muscle strengthening activities included activities to increase strength, capacity, endurance and muscle size, for example exercise with the use of bodyweight and aerobic activities. Aerobic activities should involve large muscle groups, for example walking, swimming, cycling and dancing [11]. The categorization was reviewed by an expert group. Appendix 1 gives an overview of the MET scores corresponding to the activities and the categorisation for bone- and muscle strengthening activities.”

To Appendix A we added an extra reference (to the health council report and article) and added the following information (line 374-379):

“* Based on the available data and the definition of the physical activity guidelines by the Dutch Health Council activities and sports were categorized as either muscle- and/or bone strengthening. Bone strengthening activities were defined as activities involving strength training and bearing body weight, for example jumping, walking stairs, walking, running and dancing. Muscle strengthening activities included activities to increase strength, capacity, endurance and muscle size, for example exercise with the use of bodyweight and aerobic activities. Aerobic activities should involve large muscle groups, for example walking, swimming, cycling and dancing.”

The broken weblink has been replaced.

Prevalence rates could indeed be calculated without cycling and walking. However, with the used definition of muscle strengthening activities by the Dutch Health Council we decided in collaboration with an expert group to include it.

Point 3: It was concluded that changes in strenuous activities at work, sports and walking during leisure time contributed the most to observed time trends in adherence to the Dutch physical activity guideline. From appendix A, the estimated energy expenditure of walking during leisure time was 2.7 MET. Based on the physical activity guidelines, people should engage in physical activity of moderate-to-vigorous (MVPA) intensity (>3MET). Consequently, walking during leisure time (2.7 MET) should have been excluded in the algorithm for calculating MVPA.

Response 3: We thank the reviewer for the suggestion. Walking during leisure time was indeed excluded from the MVPA trend analysis. At the same time, it was included in the bone- and muscle strengthening guideline trend analysis, as walking is classified as a bone- and muscle strengthening activity. Thereby, also included in the overall analysis of the physical activity guidelines. In Table 2 and 3, walking during leisure time is presented because the time spend on the activities in hours/week reflects the amount of time spend (per session) and the number of times (frequency). We assumed that when the total amount of time increased, the number of time had increased as well. We added the following information to the method section (line 155-157):

“The mean time spend on activities was assumed to represent duration and frequency and was presented for relevant subgroups (age, sex, level of education).”.

Point 4: Furthermore, previous work on the ‘physical activity paradox’ argues that occupational physical activity is associated with an elevated cardiovascular risk (e.g., Holtermann et al., British Journal of Sports Medicine 2012;46:291-295). This could suggest that promoting occupation physical activity as health-enhancing is not recommended. Please comment on this in the discussion part of the paper.

Response 4: We acknowledge the findings by Holtermann et al. [1]. Indeed we do not wish to promote occupational physical activity. To improve the paper we have added the following paragraph in the discussion (line 306-313):

“In our study, strenuous activities at work were found to attribute to an increasing adherence to the physical activity guidelines over time. However, it should be noted that previous research has indicated that occupational physical activity elevates cardiovascular risk [44]. In our article strenuous activities at work were seen as activities to adhere to the physical activity guidelines, as the guidelines do not make a distinction where activities have taken place [10,11]. However, bias caused by including occupational activities in the analysis might be less pronounced as our population is national representative with only a portion in strenuous labour force. Nonetheless, occupational physical activities should not be seen as activities to be promoted for health-enhancing purposes [44].”

Round 2

Reviewer 3 Report

Thank you for this revised version of the manuscript. I find that you have addressed and revised according to my comments.